# Elucidating the Preconditioning in Consistency Distillation

**Kaiwen Zheng**[†1*], **Guande He**[†1*], **Jianfei Chen**[1], **Fan Bao**[12], **Jun Zhu**[‡123]

[1]Dept. of Comp. Sci. & Tech., Institute for AI, BNRist Center, THBI Lab
[1]Tsinghua-Bosch Joint ML Center, Tsinghua University, Beijing, China
[2]Shengshu Technology, Beijing    [3]Pazhou Lab (Huangpu), Guangzhou, China
zkwthu@gmail.com; guande.he17@outlook.com;
fan.bao@shengshu.ai; {jianfeic, dcszj}@tsinghua.edu.cn

## Abstract

Consistency distillation is a prevalent way for accelerating diffusion models adopted in consistency (trajectory) models, in which a student model is trained to traverse backward on the probability flow (PF) ordinary differential equation (ODE) trajectory determined by the teacher model. Preconditioning is a vital technique for stabilizing consistency distillation, by linear combining the input data and the network output with pre-defined coefficients as the consistency function. It imposes the boundary condition of consistency functions without restricting the form and expressiveness of the neural network. However, previous preconditionings are hand-crafted and may be suboptimal choices. In this work, we offer the first theoretical insights into the preconditioning in consistency distillation, by elucidating its design criteria and the connection to the teacher ODE trajectory. Based on these analyses, we further propose a principled way dubbed *Analytic-Precond* to analytically optimize the preconditioning according to the consistency gap (defined as the gap between the teacher denoiser and the optimal student denoiser) on a generalized teacher ODE. We demonstrate that Analytic-Precond can facilitate the learning of trajectory jumpers, enhance the alignment of the student trajectory with the teacher's, and achieve $2\times$ to $3\times$ training acceleration of consistency trajectory models in multi-step generation across various datasets.

## 1 Introduction

Diffusion models are a class of powerful deep generative models, showcasing cutting-edge performance in diverse domains including image synthesis (Dhariwal & Nichol, 2021; Karras et al., 2022), speech and video generation (Chen et al., 2021; Ho et al., 2022), controllable image manipulation (Nichol et al., 2022; Ramesh et al., 2022; Rombach et al., 2022; Meng et al., 2022b), density estimation (Song et al., 2021b; Kingma et al., 2021; Lu et al., 2022a; Zheng et al., 2023b) and inverse problem solving (Chung et al., 2022; Kawar et al., 2022). Compared to their generative counterparts like variational auto-encoders (VAEs) (Kingma & Welling, 2014) and generative adversarial networks (GANs) (Goodfellow et al., 2014), diffusion models excel in high-quality generation while circumventing issues of mode collapse and training instability. Consequently, they serve as the cornerstone of next-generation generative systems like text-to-image (Rombach et al., 2022) and text-to-video (Gupta et al., 2023; Bao et al., 2024) synthesis.

The primary bottleneck for integrating diffusion models into downstream tasks lies in their slow inference processes, which gradually remove noise from data with hundreds of network evaluations. The sampling process typically involves simulating the probability flow (PF) ordinary ODE backward in time, starting from noise (Song et al., 2021c). To accelerate diffusion sampling, various training-free samplers have been proposed as specialized solvers of the PF-ODE (Song et al., 2021a; Zhang & Chen, 2022; Lu et al., 2022b), yet they still require over 10 steps to generate satisfactory samples due to the inherent discretization errors present in all numerical ODE solvers.

---

*Work done during an internship at Shengshu;    †Equal contribution;    ‡The corresponding author.

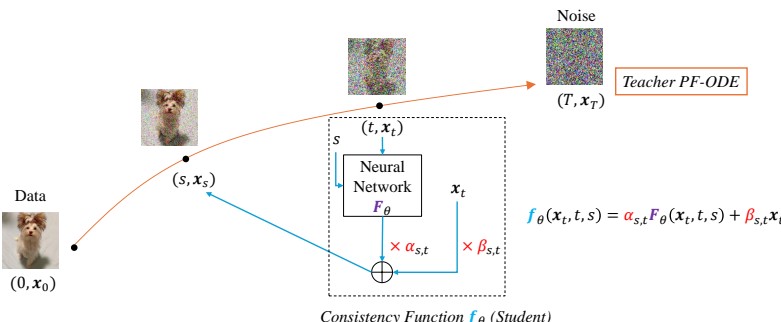

Figure 1: Consistency distillation with *preconditioning* coefficients $\alpha, \beta$.

Recent advancements in few-step or even single-step generation of diffusion models are concentrated on distillation methods (Luhman & Luhman, 2021; Salimans & Ho, 2022; Meng et al., 2022a; Song et al., 2023; Kim et al., 2023; Sauer et al., 2023). Particularly, consistency models (CMs) (Song et al., 2023) have emerged as a prominent method for diffusion distillation and successfully been applied to various data domains including latent space (Luo et al., 2023), audio (Ye et al., 2023) and video (Wang et al., 2023). CMs consider training a student network to map arbitrary points on the PF-ODE trajectory to its starting point, thereby enabling one-step generation that directly maps noise to data. A follow-up work named consistency trajectory models (CTMs) (Kim et al., 2023) extends CMs by changing the mapping destination to encompass not only the starting point but also intermediate ones, facilitating unconstrained backward jumps on the PF-ODE trajectory. This design enhances training flexibility and permits the incorporation of auxiliary losses.

In both CMs and CTMs, the mapping function (referred to as the consistency function) must adhere to certain constraints. For instance, in CMs, there exists a boundary condition dictating that the starting point maps to itself. Consequently, the consistency functions are parameterized as a linear combination of the input data and the network output with pre-defined coefficients. This approach ensures that boundary conditions are naturally satisfied without constraining the form or expressiveness of the neural network. We term this parameterization technique *preconditioning* in consistency distillation (Figure 1), aligning with the terminology in EDM (Karras et al., 2022). The preconditionings in CMs and CTMs are intuitively crafted but may be suboptimal. Besides, despite efforts, CTMs have struggled to identify any distinct preconditionings that outperform the original one.

In this work, we take the first step towards designing and enhancing preconditioning in consistency distillation to learn better "trajectory jumpers". We elucidate the design criteria of preconditioning by linking it to the discretization of the teacher ODE trajectory. We further convert the teacher PF-ODE into a generalized form involving free parameters, which induces a novel family of preconditionings. Through theoretical analyses, we unveil the significance of the consistency gap (referring to the gap between the teacher denoiser and the optimal student denoiser) in achieving good initialization and facilitating learning. By minimizing a derived bound of the consistency gap, we can optimize the preconditioning within our proposed family. We name the optimal preconditioning under our principle as *Analytic-Precond*, as it can be analytically computed according to the teacher model without manual design or hyperparameter tuning. Moreover, the computation is efficient with less than 1% time cost of the training process.

We demonstrate the effectiveness of Analytic-Precond by applying it to CMs and CTMs on standard benchmark datasets, including CIFAR-10, FFHQ 64×64 and ImageNet 64×64. While the vanilla preconditioning closely approximates Analytic-Precond and yields similar results in CMs, Analytic-Precond exhibits notable distinctions from its original counterpart in CTMs, particularly concerning intermediate jumps on the trajectory. Remarkably, Analytic-Precond achieves 2× to 3× training acceleration in CTMs in multi-step generation.

## 2 BACKGROUND

### 2.1 DIFFUSION MODELS

Diffusion models (Song et al., 2021c; Sohl-Dickstein et al., 2015; Ho et al., 2020) transform a *d*-

dimensional data distribution $q_0(\boldsymbol{x}_0)$ into Gaussian noise distribution through a forward stochastic differential equation (SDE) starting from $\boldsymbol{x}_0 \sim q_0$:

$$\mathrm{d}\boldsymbol{x}_t = f(t)\boldsymbol{x}_t\mathrm{d}t + g(t)\mathrm{d}\boldsymbol{w}_t \tag{1}$$

where $t \in [0, T]$ for some finite horizon $T$, $f, g : [0, T] \to \mathbb{R}$ is the scalar-valued drift and diffusion term, and $\boldsymbol{w}_t \in \mathbb{R}^d$ is a standard Wiener process. The forward SDE is accompanied by a series of marginal distributions $\{q_t\}_{t=0}^T$ of $\{\boldsymbol{x}_t\}_{t=0}^T$, and $f, g$ are properly designed so that the terminal distribution is approximately a pure Gaussian, i.e., $q_T(\boldsymbol{x}_T) \approx \mathcal{N}(\boldsymbol{0}, \sigma_T^2\boldsymbol{I})$. An intriguing characteristic of this SDE lies in the presence of the probability flow (PF) ODE (Song et al., 2021c) $\mathrm{d}\boldsymbol{x}_t = [f(t)\boldsymbol{x}_t - \frac{1}{2}g^2(t)\nabla_{\boldsymbol{x}_t} \log q_t(\boldsymbol{x}_t)]\mathrm{d}t$ whose solution trajectories at time $t$, when solved backward from time $T$ to time 0, are distributed exactly as $q_t$. The only unknown term $\nabla_{\boldsymbol{x}_t} \log q_t(\boldsymbol{x}_t)$ is the *score function* and can be learned by denoising score matching (DSM) (Vincent, 2011).

A prevalent noise schedule $f = 0, g = \sqrt{2t}$ is proposed by EDM (Karras et al., 2022) and followed in recent text-to-image generation (Esser et al., 2024), video generation (Blattmann et al., 2023), as well as consistency distillation. In this case, the forward transition kernel of the forward SDE (Eqn. (1)) owns a simple form $q(\boldsymbol{x}_t|\boldsymbol{x}_0) = \mathcal{N}(\boldsymbol{x}_0, t^2\boldsymbol{I})$, and the terminal distribution $q_T \approx \mathcal{N}(\boldsymbol{0}, T^2\boldsymbol{I})$. Besides, the PF-ODE can be represented by the *denoiser function* $\boldsymbol{D}_\phi(\boldsymbol{x}_t, t)$:

$$\frac{\mathrm{d}\boldsymbol{x}_t}{\mathrm{d}t} = \frac{\boldsymbol{x}_t - \boldsymbol{D}_\phi(\boldsymbol{x}_t, t)}{t} \tag{2}$$

where the denoiser function is trained to predict $\boldsymbol{x}_0$ given noisy data $\boldsymbol{x}_t = \boldsymbol{x}_0 + t\boldsymbol{\epsilon}, \boldsymbol{\epsilon} \sim \mathcal{N}(\boldsymbol{0}, \boldsymbol{I})$ at any time $t$, i.e., minimizing $\mathbb{E}_t\mathbb{E}_{p_{\text{data}}(\boldsymbol{x}_0)q(\boldsymbol{x}_t|\boldsymbol{x}_0)}[w(t)\|\boldsymbol{D}_\phi(\boldsymbol{x}_t, t) - \boldsymbol{x}_0\|_2^2]$ for some weighting $w(t)$. This denoising loss is equivalent to the DSM loss (Song et al., 2021c). In EDM, another key insight is to employ preconditioning by parameterizing $\boldsymbol{D}_\phi$ as $\boldsymbol{D}_\phi(\boldsymbol{x}, t) = c_{\text{skip}}(t)\boldsymbol{x} + c_{\text{out}}(t)\boldsymbol{F}_\phi(\boldsymbol{x}, t)$[1], where

$$c_{\text{skip}}(t) = \frac{\sigma_{\text{data}}^2}{\sigma_{\text{data}}^2 + t^2}, \quad c_{\text{out}}(t) = \frac{\sigma_{\text{data}}t}{\sqrt{\sigma_{\text{data}}^2 + t^2}}, \tag{3}$$

$\boldsymbol{F}_\phi$ is a free-form neural network and $\sigma_{\text{data}}^2$ is the variance of the data distribution.

## 2.2 CONSISTENCY DISTILLATION

Denote $\phi$ as the parameters of the teacher diffusion model, and $\theta$ as the parameters of the student network. Given a trajectory $\{\boldsymbol{x}_t\}_{t=\epsilon}^T$ with a fixed initial timestep $\epsilon$ of a teacher PF-ODE[2], consistency models (CMs) (Song et al., 2023) aim to learn a *consistency function* $\boldsymbol{f}_\theta : (\boldsymbol{x}_t, t) \mapsto \boldsymbol{x}_\epsilon$ which maps the point $\boldsymbol{x}_t$ at any time $t$ on the trajectory to the initial point $\boldsymbol{x}_\epsilon$. The consistency function is forced to satisfy the *boundary condition* $\boldsymbol{f}_\theta(\boldsymbol{x}, \epsilon) = \boldsymbol{x}$. To ensure unrestricted form and expressiveness of the neural network, $\boldsymbol{f}_\theta$ is parameterized as

$$\boldsymbol{f}_\theta(\boldsymbol{x}, t) = \frac{\sigma_{\text{data}}^2}{\sigma_{\text{data}}^2 + (t - \epsilon)^2}\boldsymbol{x} + \frac{\sigma_{\text{data}}(t - \epsilon)}{\sqrt{\sigma_{\text{data}}^2 + t^2}}\boldsymbol{F}_\theta(\boldsymbol{x}, t) \tag{4}$$

which naturally satisfies the boundary condition for any free-form network $\boldsymbol{F}_\theta(\boldsymbol{x}_t, t)$. We refer to this technique as *preconditioning* in consistency distillation, aligning with the terminology in EDM. The student $\theta$ can be distilled from the teacher by the training objective:

$$\mathbb{E}_{t\in[\epsilon,T],s\in[\epsilon,t)}\mathbb{E}_{q_0(\boldsymbol{x}_0)q(\boldsymbol{x}_t|\boldsymbol{x}_0)}\left[w(t)d\left(\boldsymbol{f}_\theta(\boldsymbol{x}_t, t), \boldsymbol{f}_{\text{sg}(\theta)}(\text{Solver}_\phi(\boldsymbol{x}_t, t, s), s)\right)\right], \tag{5}$$

where $w(\cdot)$ is a positive weighting function, $d(\cdot, \cdot)$ is a distance metric, sg is the (exponential moving average) stop-gradient and $\text{Solver}_\phi$ is any numerical solver for the teacher PF-ODE.

Consistency trajectory models (CTMs) (Kim et al., 2023) extend CMs by changing the mapping destination to not only the initial point but also any intermediate ones, enabling unconstrained backward jumps on the PF-ODE. Specifically, the consistency function is instead defined as $\boldsymbol{f}_\theta : (\boldsymbol{x}_t, t, s) \mapsto \boldsymbol{x}_s$, which maps the point $\boldsymbol{x}_t$ at time $t$ on the trajectory to the point $\boldsymbol{x}_s$ at any

---

[1]More precisely, $\boldsymbol{D}_\phi(\boldsymbol{x}, t) = c_{\text{skip}}(t)\boldsymbol{x} + c_{\text{out}}(t)\boldsymbol{F}_\phi(c_{\text{in}}(t)\boldsymbol{x}, c_{\text{noise}}(t))$. Since $c_{\text{in}}(t)$ and $c_{\text{noise}}(t)$ take effects inside the network, we absorb them into the definition of $\boldsymbol{F}_\phi$ for simplicity.

[2]In EDM, the range of timesteps is typically chosen as $\epsilon = 0.002, T = 80$.

previous time $s < t$. The boundary condition is $\boldsymbol{f}_\theta(\boldsymbol{x}, t, t) = \boldsymbol{x}$, which is forced by the following preconditioning:

$$\boldsymbol{f}_\theta(\boldsymbol{x}, t, s) = \frac{s}{t}\boldsymbol{x} + \left(1 - \frac{s}{t}\right)\boldsymbol{D}_\theta(\boldsymbol{x}, t, s) \tag{6}$$

where $\boldsymbol{D}_\theta(\boldsymbol{x}_t, t, s) = c_{\text{skip}}(t)\boldsymbol{x} + c_{\text{out}}(t)\boldsymbol{F}_\theta(\boldsymbol{x}, t, s)$ is the student denoiser function, and $\boldsymbol{F}_\theta(\boldsymbol{x}, t, s)$ is a free-form network with an extra timestep input $s$. The student network is trained by minimizing

$$\mathbb{E}_{t\in[\epsilon,T], s\in[\epsilon,t] u\in[s,t]}\mathbb{E}_{q_0(\boldsymbol{x}_0)q(\boldsymbol{x}_t|\boldsymbol{x}_0)}$$
$$\left[w(t)d\left(\boldsymbol{f}_{\text{sg}(\theta)}(\boldsymbol{f}_\theta(\boldsymbol{x}_t, t, s), s, \epsilon), \boldsymbol{f}_{\text{sg}(\theta)}(\boldsymbol{f}_{\text{sg}(\theta)}(\text{Solver}_\phi(\boldsymbol{x}_t, t, u), u, s), s, \epsilon))\right] \tag{7}$$

An important property of CTM's precondtioning is that when $s \to t$, the optimal denoiser satisfies $\boldsymbol{D}_{\theta^*}(\boldsymbol{x}, t, s) \to \boldsymbol{D}_\phi(\boldsymbol{x}_t, t)$, i.e. the diffusion denoiser. Consequently, the DSM loss in diffusion models can be incorporated to regularize the training of $\theta$, which enhances the sample quality as the number of sampling steps increases, enabling speed-quality trade-off.

## 3 METHOD

Beyond the hand-crafted preconditionings outlined in Eqn. (4) and Eqn. (6), we seek a general paradigm of preconditioning design in consistency distillation. We first analyze their key ingredients and relate them to the discretization of the teacher ODE. Then we derive a generalized ODE form, which can induce a novel family of preconditionings. Finally, we propose a principled way to analytically obtain optimized preconditioning by minimizing the consistency gap.

### 3.1 ANALYZING THE PRECONDITIONING IN CONSISTENCY DISTILLATION

We examine the form of consistency function $\boldsymbol{f}_\theta(\boldsymbol{x}, t, s)$ in CTMs, wherein it subsumes CMs as a special case by setting the jumping destination $s$ as the initial timestep $\epsilon$. Assume $\boldsymbol{f}_\theta$ is parameterized as the following form of skip connection:

$$\boldsymbol{f}_\theta(\boldsymbol{x}, t, s) = f(t, s)\boldsymbol{x} + g(t, s)\boldsymbol{D}_\theta(\boldsymbol{x}, t, s) \tag{8}$$

where $\boldsymbol{D}_\theta(\boldsymbol{x}, t, s) = c_{\text{skip}}(t)\boldsymbol{x} + c_{\text{out}}(t)\boldsymbol{F}_\theta(\boldsymbol{x}, t, s)$ represents the student denoiser function in alignment with EDM, and $f(t, s), g(t, s)$ are coefficients that linearly combine $\boldsymbol{x}$ and $\boldsymbol{D}_\theta$. We identify two essential constraints on the coefficients $f$ and $g$.

**Boundary Condition**   For any free-form network $\boldsymbol{F}_\theta$ or $\boldsymbol{D}_\theta$, the consistency function $\boldsymbol{f}_\theta$ must adhere to $\boldsymbol{f}_\theta(\boldsymbol{x}, t, t) = \boldsymbol{x}$ (in-place jumping retains the original data point). Therefore, $f$ and $g$ should meet the conditions $f(t, t) = 1$ and $g(t, t) = 0$ for any time $t$.

**Alignment with the Denoiser**   Denote the optimal consistency function that precisely follows the teacher PF-ODE trajectory as $\boldsymbol{f}_{\theta^*}(\boldsymbol{x}, t, s)$, and the optimal denoiser as $\boldsymbol{D}_{\theta^*}(\boldsymbol{x}, t, s) = \frac{\boldsymbol{f}_{\theta^*}(\boldsymbol{x}, t, s) - f(t, s)\boldsymbol{x}}{g(t, s)}$ according to Eqn. (8). In CTMs, $f$ and $g$ are properly designed so that the limit $\boldsymbol{D}_{\theta^*}(\boldsymbol{x}, t, t) = \lim_{s\to t}\frac{\boldsymbol{f}_{\theta^*}(\boldsymbol{x}, t, s) - f(t, s)\boldsymbol{x}}{g(t, s)} = \boldsymbol{D}_\phi(\boldsymbol{x}, t)$. Thus, the student denoiser at $s = t$, i.e. $\boldsymbol{D}_\theta(\boldsymbol{x}, t, t)$, ideally aligns with the teacher denoiser $\boldsymbol{D}_\phi$. This alignment offers two advantages: (1) $\boldsymbol{D}_\theta(\boldsymbol{x}_t, t, t)$ acts as a valid diffusion denoiser and is amenable to regularization with the DSM loss. (2) The teacher model $\boldsymbol{D}_\phi$ serves as an effective initializer of the student $\boldsymbol{D}_\theta$ at $s = t$, implying that $\boldsymbol{D}_\theta$ solely at $s < t$ is suboptimal and requires further optimization.

Precondionings satisfying these constraints can be derived by discretizing the teacher PF-ODE. Suppose the discretization from time $t$ to time $s$ is expressed as $\boldsymbol{x}_s = f(t, s)\boldsymbol{x}_t + g(t, s)\boldsymbol{D}_\phi(\boldsymbol{x}_t, t)$, then $f, g$ naturally satisfy the conditions: the discretization from $t$ to $t$ must be $\boldsymbol{x}_t = \boldsymbol{x}_t$; as $s \to t$, the discretization error tends to 0, and the optimal student for conducting infinitesimally small jumps is just $\boldsymbol{D}_\phi(\boldsymbol{x}_t, t)$. For instance, applying Euler method to the PF-ODE in Eqn. (2) yields:

$$\boldsymbol{x}_s - \boldsymbol{x}_t = (s - t)\frac{\boldsymbol{x}_t - \boldsymbol{D}_\phi(\boldsymbol{x}_t, t)}{t} \Rightarrow \boldsymbol{x}_s = \frac{s}{t}\boldsymbol{x}_t + \left(1 - \frac{s}{t}\right)\boldsymbol{D}_\phi(\boldsymbol{x}_t, t) \tag{9}$$

which exactly matches the preconditioning used in CTMs by replacing $\boldsymbol{D}_\phi(\boldsymbol{x}_t, t)$ with $\boldsymbol{D}_\theta(\boldsymbol{x}_t, t, s)$. Elucidating preconditioning as ODE discretization also closely approximates CMs' choice in

Eqn. (4). For $t \gg \epsilon$, we have $t - \epsilon \approx t$, therefore $\boldsymbol{f}_\theta$ in Eqn. (4) approximately equals the denoiser $\boldsymbol{D}_\theta$. On the other hand, as $\frac{\epsilon}{t} \approx 0$, CTMs' choice in Eqn. (6) also indicates $\boldsymbol{f}_\theta \approx \boldsymbol{D}_\theta$. Therefore, CMs' preconditioning is only distinct from ODE discretization when $t$ is close to $\epsilon$, which is not the case in one-step or few-step generation.

## 3.2 INDUCED PRECONDITIONING BY GENERALIZED ODE

Based on the analyses above, the preconditioning can be induced from ODE discretization. Drawing inspirations from the dedicated ODE solvers in diffusion models (Lu et al., 2022b; Zheng et al., 2023a), we consider a generalized representation of the teacher ODE in Eqn. (2), which can give rise to alternative preconditionings that satisfy the restrictions.

Firstly, we modulate the ODE with a continuous function $L_t$ to transform it into an ODE with respect to $L_t \boldsymbol{x}_t$ rather than $\boldsymbol{x}_t$. Leveraging the chain rule of derivatives, we obtain $\frac{\mathrm{d}(L_t \boldsymbol{x}_t)}{\mathrm{d}t} = L_t \frac{\mathrm{d}\boldsymbol{x}_t}{\mathrm{d}t} + \frac{\mathrm{d}L_t}{\mathrm{d}t} \boldsymbol{x}_t$, where $\frac{\mathrm{d}\boldsymbol{x}_t}{\mathrm{d}t}$ can be substituted by the original teacher ODE, resulting in

$$\frac{\mathrm{d}(L_t \boldsymbol{x}_t)}{\mathrm{d}t} = \frac{L_t}{t} \left[ \left( 1 + \frac{\mathrm{d} \log L_t}{\mathrm{d}t} t \right) \boldsymbol{x}_t - \boldsymbol{D}_\phi(\boldsymbol{x}_t, t) \right] \tag{10}$$

By changing the time variable from $t$ to $\lambda_t = -\log t$, the ODE can be further simplified to

$$\frac{\mathrm{d}(L_t \boldsymbol{x}_t)}{\mathrm{d}\lambda_t} = L_t \boldsymbol{g}_\phi(\boldsymbol{x}_t, t), \quad \boldsymbol{g}_\phi(\boldsymbol{x}_t, t) := \boldsymbol{D}_\phi(\boldsymbol{x}_t, t) - (1 - l_t)\boldsymbol{x}_t \tag{11}$$

where we denote $l_t := \frac{\mathrm{d} \log L_{t_\lambda}}{\mathrm{d}\lambda}$, and $t_\lambda = e^{-\lambda}$ is the inverse function of $\lambda_t$. Moreover, $L_t$ can be represented by $l_t$ as $L_t = e^{\int_{\lambda_T}^{\lambda_t} l_{t_\lambda} \mathrm{d}\lambda}$. Secondly, instead of using $t$ or $\lambda_t$ as the time variable in the ODE (i.e., formulate the ODE as $\frac{\mathrm{d}(\cdot)}{\mathrm{d}t}$ or $\frac{\mathrm{d}(\cdot)}{\mathrm{d}\lambda_t}$), we can employ a generalized time representation $\eta_t = \int_{\lambda_T}^{\lambda_t} L_{t_\lambda} S_{t_\lambda} \mathrm{d}\lambda$, where $S_t$ is any positive continuous function. This transformation ensures that $\eta$ monotonically increases with respect to $\lambda$, enabling one-to-one inverse mappings $t_\eta, \lambda_\eta$. To align with $L_t$, we express $S_t$ as $e^{\int_{\lambda_T}^{\lambda_t} s_{t_\lambda} \mathrm{d}\lambda}$, where we denote $s_t := \frac{\mathrm{d} \log S_{t_\lambda}}{\mathrm{d}\lambda}$. Using $\eta_t$ as the new time variable, we have $\mathrm{d}\eta = L_{t_\lambda} S_{t_\lambda} \mathrm{d}\lambda$, and the ODE in Eqn. (11) is further generalized to

$$\frac{\mathrm{d}(L_t \boldsymbol{x}_t)}{\mathrm{d}\eta_t} = \frac{\boldsymbol{g}_\phi(\boldsymbol{x}_t, t)}{S_t} \tag{12}$$

The final generalized ODE in Eqn. (12) is theoretically equivalent to the original teacher PF-ODE in Eqn. (2), albeit with a set of introduced free parameters $\{l_t, s_t\}_{t=\epsilon}^T$. Applying the Euler method leads to different discretizations from Eqn. (9):

$$L_s \boldsymbol{x}_s - L_t \boldsymbol{x}_t = (\eta_s - \eta_t) \frac{\boldsymbol{g}_\phi(\boldsymbol{x}_t, t)}{S_t} \tag{13}$$

which can be rearranged as

$$\boldsymbol{x}_s = \frac{L_t S_t + (l_t - 1)(\eta_s - \eta_t)}{L_s S_t} \boldsymbol{x}_t + \frac{\eta_s - \eta_t}{L_s S_t} \boldsymbol{D}_\phi(\boldsymbol{x}_t, t) \tag{14}$$

Hence, the induced preconditioning can be expressed by Eqn. (8) with a novel set of coefficients $f(t, s) = \frac{L_t S_t + (l_t - 1)(\eta_s - \eta_t)}{L_s S_t}, g(t, s) = \frac{\eta_s - \eta_t}{L_s S_t}$. Originating from the Euler discretization of an equivalent teacher ODE, these coefficients adhere to the constraints outlined in Section 3.1 under any parameters $\{l_t, s_t\}_{t=\epsilon}^T$, thus opening avenues for further optimization. The induced preconditioning can also degenerate to CTM's case $f(t, s) = \frac{s}{t}, g(t, s) = 1 - \frac{s}{t}$ under specific selections $l_t = 0, s_t = -1$ for $t \in [\epsilon, T]$.

## 3.3 PRINCIPLES FOR OPTIMIZING THE PRECONDITIONING

Derived from the generalized teacher ODE presented in Eqn. (12), a range of preconditionings is now at our disposal with coefficients $f, g$ from Eqn. (14), governed by the free parameters $\{l_t, s_t\}_{t=\epsilon}^T$. Our aim is to establish guiding principles for discerning the optimal sets of $\{l_t, s_t\}_{t=\epsilon}^T$, thereby attaining superior preconditioning compared to the original one in Eqn. (6).

Table 1: Comparison between different preconditionings used in consistency distillation.

| Method | CM (Song et al., 2023) | BCM (Li & He, 2024) | CTM (Kim et al., 2023) | Analytic-Precond (**Ours**) |
|---|---|---|---|---|
| Free-form Network | $\boldsymbol{F}_\theta(\boldsymbol{x}_t, t)$ | $\boldsymbol{F}_\theta(\boldsymbol{x}_t, t, s)$ | | |
| Denoiser Function | $\boldsymbol{D}_\theta(\boldsymbol{x}, t) = c_{\text{skip}}(t)\boldsymbol{x}$ $+ c_{\text{out}}(t)\boldsymbol{F}_\theta(\boldsymbol{x}, t)$ | $\boldsymbol{D}_\theta(\boldsymbol{x}, t, s) = c_{\text{skip}}(t)\boldsymbol{x} + c_{\text{out}}(t)\boldsymbol{F}_\theta(\boldsymbol{x}, t, s)$ | | |
| Consistency Function | $\boldsymbol{f}_\theta(\boldsymbol{x}, t) = f(t, \epsilon)\boldsymbol{x}$ $+ g(t, \epsilon)\boldsymbol{F}_\theta(\boldsymbol{x}, t)$ | $\boldsymbol{f}_\theta(\boldsymbol{x}, t, s) = f(t, s)\boldsymbol{x}$ $+ g(t, s)\boldsymbol{F}_\theta(\boldsymbol{x}, t, s)$ | $\boldsymbol{f}_\theta(\boldsymbol{x}, t, s) = f(t, s)\boldsymbol{x}$ $+ g(t, s)\boldsymbol{D}_\theta(\boldsymbol{x}, t, s)$ | |
| $f(t, s)$ | $\dfrac{\sigma_{\text{data}}^2}{\sigma_{\text{data}}^2 + (t - s)^2}$ | $\dfrac{\sigma_{\text{data}}^2 + ts}{\sigma_{\text{data}}^2 + t^2}$ | $\dfrac{s}{t}$ | $\dfrac{L_t S_s}{L_s S_s + (1 - l_s)(\eta_s - \eta_t)}$ |
| $g(t, s)$ | $\dfrac{\sigma_{\text{data}}(t - s)}{\sqrt{\sigma_{\text{data}}^2 + t^2}}$ | $\dfrac{\sigma_{\text{data}}(t - s)}{\sqrt{\sigma_{\text{data}}^2 + t^2}}$ | $1 - \dfrac{s}{t}$ | $\dfrac{\eta_s - \eta_t}{L_s S_s + (1 - l_s)(\eta_s - \eta_t)}$ |

Firstly, drawing from the insights of Rosenbrock-type exponential integrators and their relevance in diffusion models (Hochbruck & Ostermann, 2010; Hochbruck et al., 2009; Zheng et al., 2023a), it is suggested that the parameter $l_t$ be chosen to restrict the gradient of Eqn. (12)'s right-hand side term with respect to $\boldsymbol{x}_t$. This choice ensures the robustness of the resulting ODE against errors in $\boldsymbol{x}_t$. An analytical solution for $l_t$ is derived as follows:

$$l_t = \operatorname*{argmin}_l \mathbb{E}_{q(\boldsymbol{x}_t)}\left[\|\nabla_{\boldsymbol{x}_t}\boldsymbol{g}_\phi(\boldsymbol{x}_t, t)\|_F\right] \Rightarrow l_t = 1 - \frac{\mathbb{E}_{q(\boldsymbol{x}_t)}\left[\operatorname{tr}(\nabla_{\boldsymbol{x}_t}\boldsymbol{D}_\phi(\boldsymbol{x}_t, t))\right]}{d} \tag{15}$$

where $d$ is the data dimensionality, $\|\cdot\|_F$ denotes the Frobenius norm and $\operatorname{tr}(\cdot)$ represents the trace of a matrix. Secondly, to determine the optimal value of $s_t$, we dive deeper into the relationship between the teacher denoiser $\boldsymbol{D}_\phi(\boldsymbol{x}_t, t)$ and the student denoiser $\boldsymbol{D}_\theta(\boldsymbol{x}_t, t, s)$. As elucidated in Section 3.1, the preconditioning is properly crafted to ensure that the optimal student denoiser satisfies $\boldsymbol{D}_{\theta^*}(\boldsymbol{x}_t, t, t) = \boldsymbol{D}_\phi(\boldsymbol{x}_t, t)$. We further explore the scenario where $s < t$ by examining the gap $\|\boldsymbol{D}_{\theta^*}(\boldsymbol{x}_t, t, s) - \boldsymbol{D}_\phi(\boldsymbol{x}_t, t)\|_2$, which we refer to as the *consistency gap*. Minimizing this gap extends the alignment of $\boldsymbol{D}_\phi$ and $\boldsymbol{D}_{\theta^*}$ to cases where $s < t$, ensuring that the teacher denoiser also serves as a good trajectory jumper. In the subsequent proposition, we derive a bound depicting the asymptotic behavior of the consistency gap:

**Proposition 3.1** (Bound for the Consistency Gap, proof in Appendix A.1). *Suppose there exists some constant $C > 0$ so that the parameters $\{l_t, s_t\}_{t=\epsilon}^T$ are bounded by $|l_t|, |s_t| \leq C$, then the optimal student denoiser function $\boldsymbol{D}_{\theta^*}$ under the preconditioning $f(t, s) = \frac{L_t S_t + (l_t - 1)(\eta_s - \eta_t)}{L_s S_t}, g(t, s) = \frac{\eta_s - \eta_t}{L_s S_t}$ satisfies*

$$\|\boldsymbol{D}_{\theta^*}(\boldsymbol{x}_t, t, s) - \boldsymbol{D}_\phi(\boldsymbol{x}_t, t)\|_2 \leq \frac{(t/s)^{3C} - 1}{3C} \max_{s \leq \tau \leq t}\left\|\frac{\mathrm{d}\boldsymbol{g}_\phi(\boldsymbol{x}_\tau, \tau)}{\mathrm{d}\lambda_\tau} - s_\tau \boldsymbol{g}_\phi(\boldsymbol{x}_\tau, \tau)\right\|_2 \tag{16}$$

The proposition conforms to the constraint $\boldsymbol{D}_{\theta^*}(\boldsymbol{x}_t, t, t) = \boldsymbol{D}_\phi(\boldsymbol{x}_t, t)$ when $s = t$. Moreover, considering $s$ in a local neighborhood of $t$, by Taylor expansion we have $\frac{(t/s)^{3C} - 1}{3C} = \frac{e^{3C(\log t - \log s)} - 1}{3C} = 1 + \mathcal{O}(\log t - \log s)$. Therefore, the consistency gap for $s \in (t - \delta, t)$, when $\delta$ is small, is roughly $\max_{s \leq \tau \leq t}\|\frac{\boldsymbol{g}_\phi(\boldsymbol{x}_\tau, \tau)}{\mathrm{d}\lambda_\tau} - s_\tau \boldsymbol{g}_\phi(\boldsymbol{x}_\tau, \tau)\|_2$. Minimizing this yields an analytic solution for $s_t$:

$$s_t = \operatorname*{argmin}_s \mathbb{E}_{q(\boldsymbol{x}_t)}\left[\left\|\frac{\mathrm{d}\boldsymbol{g}_\phi(\boldsymbol{x}_t, t)}{\mathrm{d}\lambda_t} - s_t \boldsymbol{g}_\phi(\boldsymbol{x}_t, t)\right\|_2\right] \Rightarrow s_t = \frac{\mathbb{E}_{q(\boldsymbol{x}_t)}\left[\boldsymbol{g}_\phi(\boldsymbol{x}_t, t)^\top \frac{\mathrm{d}\boldsymbol{g}_\phi(\boldsymbol{x}_t, t)}{\mathrm{d}\lambda_t}\right]}{\mathbb{E}_{q(\boldsymbol{x}_t)}\left[\|\boldsymbol{g}_\phi(\boldsymbol{x}_t, t)\|_2^2\right]} \tag{17}$$

We term the resulting preconditioning as *Analytic-Precond*, as $l_t, s_t$ are analytically determined by the teacher $\phi$ using Eqn. (15) and Eqn. (17). Though $l_t, s_t$ are defined over continuous timesteps, we can compute them on hundreds of discretized ones, while obtaining reasonable estimations of their related terms $L_t, S_t, \eta_t$. The computation is highly efficient utilizing automatic differentiation in modern deep learning frameworks, requiring less than 1% of the total training time (Appendix B.1).

**Backward Euler Method for Training Stability**   Despite the approximation $\frac{(t/s)^{3C} - 1}{3C} \approx 1$ holding true in local neighborhoods of $t$, the coefficient $\frac{(t/s)^{3C} - 1}{3C}$ in the bound exhibits exponential

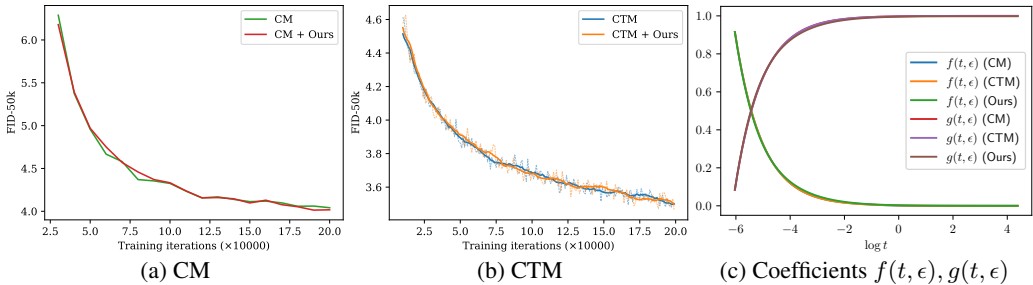

Figure 2: Training curves for single-step generation, and visualization of preconditionings for single-step jump on CIFAR-10 (conditional).

behavior when $\frac{t}{s} \gg 1$. In practice, directly applying the preconditioning derived from Eqn. (14) may cause training instability, especially on long jumps with large step sizes. Drawing inspiration from the stability of the backward Euler method, known for its efficacy in handling stiff equations without step size restrictions, we propose a backward rewriting of Eqn. (14) from $s$ to $t$ as $\boldsymbol{x}_t = \hat{f}(s,t)\boldsymbol{x}_s + \hat{g}(s,t)\boldsymbol{D}_\phi$, where $\hat{f}, \hat{g}$ are the original coefficients from Eqn. (14). Rearranging this equation yields $\boldsymbol{x}_s = \frac{1}{\hat{f}(s,t)}\boldsymbol{x}_t - \frac{\hat{g}(s,t)}{\hat{f}(s,t)}\boldsymbol{D}_\phi$, giving rise to the *backward coefficients* $f(t,s) = \frac{1}{\hat{f}(s,t)}, g(t,s) = -\frac{\hat{g}(s,t)}{\hat{f}(s,t)}$.

We summarize different preconditionings in Table 1, where we also included a concurrent work called bidirectional consistency models (BCMs) (Li & He, 2024) which proposed an alternative preconditioning to CTMs.

## 4 RELATED WORK

**Fast Diffusion Sampling** Fast sampling of diffusion models can be categorized into training-free and training-based methods. The former typically seek implicit sampling processes (Song et al., 2021a; Zheng et al., 2024a;b) or dedicated numerical solvers to the differential equations corresponding to diffusion generation, including Heun's methods (Karras et al., 2022), splitting numerical methods (Wizadwongsa & Suwajanakorn, 2022), pseudo numerical methods (Liu et al., 2021) and exponential integrators (Zhang & Chen, 2022; Lu et al., 2022b; Zheng et al., 2023a; Gonzalez et al., 2023). They typically require around 10 steps for high-quality generation. In contrast, training-based methods, particularly adversarial distillation (Sauer et al., 2023) and consistency distillation (Song et al., 2023; Kim et al., 2023), have gained prominence for their ability to achieve high-quality generation with just one or two steps. While adversarial distillation proves its effectiveness in one-step generation of text-to-image diffusion models (Sauer et al., 2024), it is theoretically less transparent than consistency distillation due to its reliance on adversarial training. Diffusion models can also be accelerated using quantized or sparse attention (Zhang et al., 2025a; 2024; 2025b).

**Parameterization in Diffusion Models** Parameterization is vital in efficient training and sampling of diffusion models. Initially, the practice involved parameterizing a noise prediction network (Ho et al., 2020; Song et al., 2021c), which outperformed direct data prediction. A notable subsequent enhancement is the introduction of "v" prediction (Salimans & Ho, 2022), which predicts the velocity along the diffusion trajectory, and is proven effective in applications like text-to-image generation (Esser et al., 2024) and density estimation (Zheng et al., 2023b). EDM (Karras et al., 2022) further advances the field by proposing a preconditioning technique that expresses the denoiser function as a linear combination of data and network, yielding state-of-the-art sample quality alongside other techniques. However, the parameterization in consistency distillation remains unexplored.

## 5 EXPERIMENTS

In this section, we demonstrate the impact of Analytic-Precond when applied to consistency distillation. Our experiments encompass various image datasets, including CIFAR-10 (Krizhevsky, 2009), FFHQ (Karras et al., 2019) 64×64, and ImageNet (Deng et al., 2009) 64×64, under both uncon-

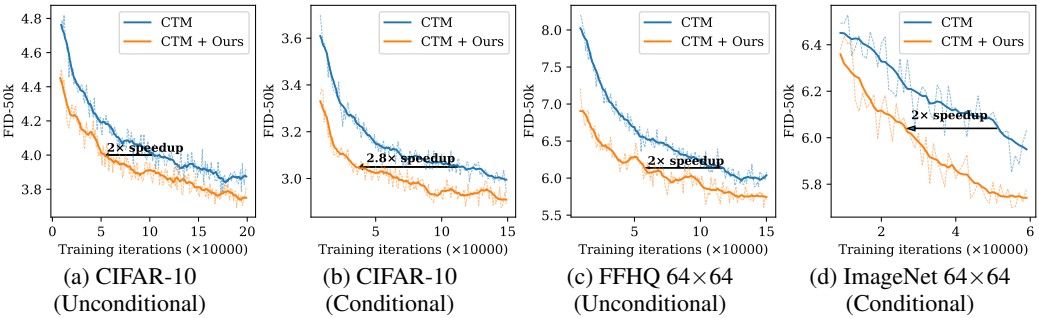

Figure 3: Training curves for two-step generation.

ditional and class-conditional settings. We deploy Analytic-Precond across two paradigms: consistency models (CMs) (Song et al., 2023) and consistency trajectory models (CTMs) (Kim et al., 2023), wherein we solely substitute the preconditioning while retaining other training procedures. For further experiment details, please refer to Appendix B.

Our investigation aims to address two primary questions:

- Can Analytic-Precond yield improvements over the original preconditioning of CMs and CTMs, across both single-step and multi-step generation?
- How does Analytic-Precond differ from prior preconditioning across datasets, concerning the coefficients $f(t, s)$ and $g(t, s)$?

## 5.1 TRAINING ACCELERATION

**Effects on CMs and Single-Step CTMs** We first apply Analytic-Precond to CMs, where the consistency function $\boldsymbol{f}_\theta(\boldsymbol{x}_t, t)$ is defined to map $\boldsymbol{x}_t$ on the teacher ODE trajectory to the starting point $\boldsymbol{x}_\epsilon$ at fixed time $\epsilon$. The models are trained with the consistency loss defined in Eqn. (5) and on the CIFAR-10 dataset, with class labels as conditions. As depicted in Figure 2 (a), we observe that Analytic-Precond yields training curves similar to original CM, measured by FID. Since multi-step consistency sampling in CMs only involves evaluating $\boldsymbol{f}_\theta(\boldsymbol{x}_t, t)$ multiple times, the results remain comparable even with an increase in sampling steps. Similar phenomena emerge in CTMs with single-step generation, as illustrated in Figure 2 (b). The commonality between these two scenarios lies in the utilization of only the jumping destination at $\epsilon$. To investigate further, we plot the preconditioning coefficients $f(t, \epsilon)$ and $g(t, \epsilon)$ in CMs, CTMs and Analytic-Precond as a function of $\log t$, as illustrated in Figure 2 (c). It is evident that across varying $t$, different preconditioning coefficients $f$ and $g$ exhibit negligible discrepancies when $s$ is fixed to $\epsilon$. This elucidates the rationale behind the comparable performance, suggesting that the original preconditionings for $t \to \epsilon$ are already quite optimal with minimal room for further optimization.

**Effects on Two-Step CTMs** We further track sample quality during the training process on CTMs, particularly focusing on two-step generation where an intermediate jump is involved ($T \to t_0 \to \epsilon$). The models are training with both the consistency trajectory loss in Eqn. (7) and the denoising score matching (DSM) loss $\mathbb{E}_t \mathbb{E}_{p_{\text{data}}(\boldsymbol{x}_0) q(\boldsymbol{x}_t | \boldsymbol{x}_0)} [w(t) \| \boldsymbol{D}_\theta(\boldsymbol{x}_t, t, t) - \boldsymbol{x}_0 \|_2^2]$, following CTMs[3]. As shown in Figure 3, across diverse datasets, Analytic-Precond enjoys superior initialization and up to $3\times$ training acceleration compared to CTM's preconditioning. This observation indicates the suboptimality of the original intermediate trajectory jumps $t \to s > \epsilon$. We provided the generated samples in Appendix C.

## 5.2 GENERATION WITH MORE STEPS

Apart from the superiority of CTMs over CMs in single-step generation (Figure 2), another notable advantage of CTMs is the regularization effects of the DSM loss. This ensures that $\boldsymbol{D}_\theta(\boldsymbol{x}_t, t, t)$ functions as a valid denoiser in diffusion models, facilitating sample quality enhancement with additional sampling steps. To evaluate the effectiveness of Analytic-Precond with more steps, we

---

[3]CTMs also propose to combine the GAN loss for further enhancing quality, which we will discuss later.

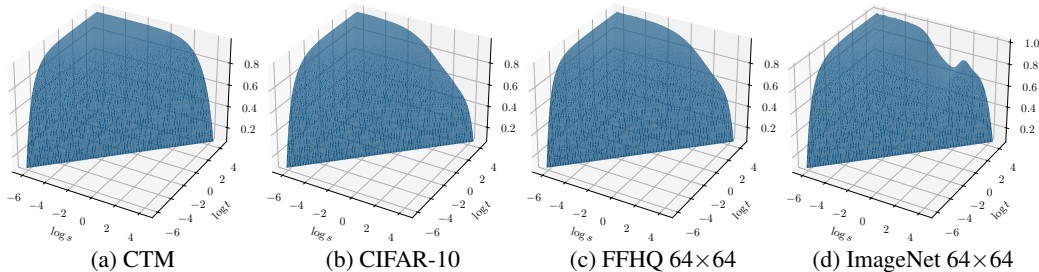

| (a) CTM | (b) CIFAR-10 | (c) FFHQ $64\times64$ | (d) ImageNet $64\times64$ |

Figure 4: Visualizations of the preconditioning coefficient $g(t,s)$ for CTM, and for Analytic-Precond under different datasets.

Table 2: FID results in multi-step generation with different number of function evaluations (NFEs).

| FID | NFE | | | | | NFE | | | | |
|---|---|---|---|---|---|---|---|---|---|---|
| | 2 | 3 | 5 | 8 | 10 | 2 | 3 | 5 | 8 | 10 |
| | CIFAR-10 (Unconditional) | | | | | CIFAR-10 (Conditional) | | | | |
| CTM | 3.83 | 3.58 | 3.43 | 3.33 | 3.22 | 3.00 | 2.82 | 2.59 | 2.67 | 2.56 |
| CTM + Ours | 3.77 | 3.54 | 3.38 | 3.30 | 3.25 | 2.92 | 2.75 | 2.62 | 2.60 | 2.65 |
| | FFHQ $64 \times 64$ (Unconditional) | | | | | ImageNet $256 \times 256$ (Conditional) | | | | |
| CTM | 5.96 | 5.80 | 5.53 | 5.39 | 5.23 | 5.95 | 6.16 | 5.43 | 5.44 | 5.98 |
| CTM + Ours | 5.71 | 5.56 | 5.47 | 5.31 | 5.12 | 5.73 | 5.67 | 5.34 | 5.43 | 5.70 |

employ the deterministic procedure in CTMs, which employs the consistency function to jump on consecutively decreasing timesteps from $T$ to $\epsilon$. As shown in Table 2, Analytic-Precond brings consistent improvement over CTMs as the number of steps increases, indicating better alignment between the consistency function and the denoiser function.

## 5.3 ANALYSES AND DISCUSSIONS

**Visualizations** To intuitively understand the distinctions between Analytic-Precond and the original preconditioning in CTMs, we investigate the variations in coefficients $f(t,s), g(t,s)$. We find that Analytic-Precond yields $f(t,s)$ close to that of CTMs, denoted as $f^{\text{CTM}}(t,s)$, with $|f^{\text{CTM}}(t,s) - f(t,s)| < 0.03$ across various $t$ and $s$. However, $g(t,s)$ produced by Analytic-Precond tends to be smaller, with disparities of up to 0.25 compared to $g^{\text{CTM}}(t,s)$. This distinction is visually demonstrated in Figure 4, where we depict $g(t,s)$ as a binary function of $\log t$ and $\log s$. Notably, the distinction is more pronounced for short jumps $t \to s$ where $|t-s|/t$ is small.

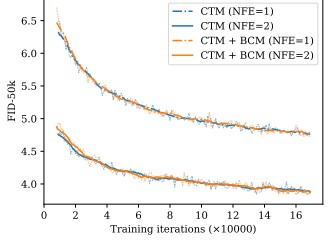

Figure 5: Effects of BCM's preconditioning on CTMs.

**Comparison to BCMs** In a concurrent work called bidirectional consistency models (BCMs) (Li & He, 2024), a novel preconditioning is derived from EDM's first principle (specified in Table 1). BCM's preconditioning also accommodates flexible transitions from $t$ to $s$ along the trajectory. However, as shown in Figure 5, replacing CTM's preconditioning with BCM's fails to bring improvements in both one-step and two-step generation.

**Compatibility with GAN loss** CTMs introduce GAN loss to further enhance the one-step generation quality, employing a discriminator and adopting an alternative optimization approach akin to GANs. As shown in Figure 6, when GAN loss is incorporated on CIFAR-10, Analytic-Precond demonstrates comparable performance. However, in this scenario, the consistency function no longer faithfully adheres to the teacher ODE trajectory, and one-step generation is even better than two-step, deviating from our theoretical foundations. Nevertheless, the utilization of Analytic-Precond does not lead to performance degradation.

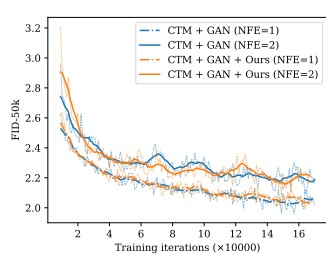

Figure 6: Effects of Analytic-Precond with GAN loss.

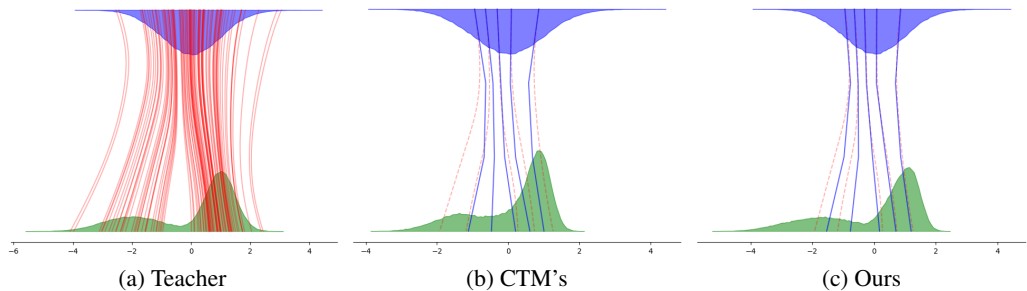

Figure 7: Visualizations of the trajectory alignment, comparing teacher and 3-step student.

**Enhancement of the Trajectory Alignment** We observe that our method also leads to lower mean square error (MSE) in the multi-step generation of CTM, when compared to the teacher diffusion model under the same initial noise, indicating enhanced fidelity to the teacher's trajectory. To better illustrate the effect of Analytic-Precond in improving trajectory alignment, we adopt a toy example where the data distribution is a simple 1-D Gaussian mixture $\frac{1}{3}\mathcal{N}(-2, 1) + \frac{2}{3}\mathcal{N}(1, 0.25)$. In this case, we can analytically derive the optimal denoiser and visualize the ground-truth teacher trajectory. We initialize the consistency function with the optimal denoiser and apply different preconditionings. As shown in Figure 7, our preconditioning produces few-step trajectories that better align with the teacher's and yields a more accurate final distribution.

## 6 CONCLUSION

In this work, we elucidate the design criteria of the preconditioning in consistency distillation for the first time and propose a novel and principled preconditioning that accelerates the training of CTMs in multi-step generation by $2\times$ to $3\times$. The crux of our approach lies in our theoretical insights, connecting preconditioning to ODE discretization, and emphasizing the alignment between the consistency function and the denoiser function. Minimizing the consistency gap fosters coordination between the consistency loss and the denoising score-matching loss, thereby facilitating speed-quality trade-offs. Our method provides the first guidelines for designing improved trajectory jumpers on the diffusion ODE, with potential applications in other types of ODE trajectories such as the dynamics of control systems or robotic path planning.

**Limitations and Broader Impact** Despite notable training acceleration in multi-step generation, the final FID improvement is relatively insignificant. Besides, Analytic-Precond fails to differ from previous preconditionings on long jumps, resulting in comparable performance in single-step generation. Achieving accelerated distillation in generative modeling may also raise concerns about the potential misuse for generating fake and malicious media content. Furthermore, it may amplify undesirable social bias that could already exist in the training dataset.

## ACKNOWLEDGMENTS

This work was supported by the National Natural Science Foundation of China (Nos. 62350080, 62106120, 92270001), Tsinghua Institute for Guo Qiang, and the High Performance Computing Center, Tsinghua University; J. Zhu was also supported by the XPlorer Prize.

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

# A    PROOFS

## A.1    PROOF OF PROPOSITION 3.1

*Proof.* Denote $\{\boldsymbol{x}_\tau\}_{\tau=s}^{t}$ as data points on the same teacher ODE trajectory. The generalized ODE in Eqn. (12) can be reformulated as an integral:

$$L_s\boldsymbol{x}_s - L_t\boldsymbol{x}_t = \int_{\eta_t}^{\eta_s} \boldsymbol{h}_\phi(\boldsymbol{x}_{t_{\lambda_\eta}}, t_{\lambda_\eta})\mathrm{d}\eta \tag{18}$$

where $\boldsymbol{h}_\phi(\boldsymbol{x}_t, t) := \frac{\boldsymbol{g}_\phi(\boldsymbol{x}_t, t)}{S_t}$, and $\boldsymbol{g}_\phi$ is defined by the teacher denoiser $\boldsymbol{D}_\phi$ in Eqn. (11). On the other hand, by replacing the teacher denoiser $\boldsymbol{D}_\phi$ with the student denoiser $\boldsymbol{D}_\theta$ in the Euler discretization (Eqn. (13)), the optimal student $\theta^*$ should satisfy

$$L_s\boldsymbol{x}_s - L_t\boldsymbol{x}_t = (\eta_s - \eta_t)\boldsymbol{h}_{\theta^*}(\boldsymbol{x}_t, t, s) \tag{19}$$

where $\boldsymbol{h}_\theta, \boldsymbol{g}_\theta$ are defined similarly to $\boldsymbol{h}_\phi, \boldsymbol{g}_\phi$ as

$$\boldsymbol{h}_\theta(\boldsymbol{x}_t, t, s) = \frac{\boldsymbol{g}_\theta(\boldsymbol{x}_t, t, s)}{S_t}, \quad \boldsymbol{g}_\theta(\boldsymbol{x}_t, t, s) = \boldsymbol{D}_\theta(\boldsymbol{x}_t, t, s) - (1 - l_t)\boldsymbol{x}_t \tag{20}$$

Combining the above equations, we have

$$\begin{aligned}
\boldsymbol{D}_{\theta^*}(\boldsymbol{x}_t, t, s) - \boldsymbol{D}_\phi(\boldsymbol{x}_t, t) &= S_t(\boldsymbol{h}_{\theta^*}(\boldsymbol{x}_t, t, s) - \boldsymbol{h}_\phi(\boldsymbol{x}_t, t)) \\
&= S_t\left(\frac{L_s\boldsymbol{x}_s - L_t\boldsymbol{x}_t}{\eta_s - \eta_t} - \boldsymbol{h}_\phi(\boldsymbol{x}_t, t)\right) \\
&= \frac{S_t}{\eta_s - \eta_t}\int_{\eta_t}^{\eta_s} \boldsymbol{h}_\phi(\boldsymbol{x}_{t_{\lambda_\eta}}, t_{\lambda_\eta}) - \boldsymbol{h}_\phi(\boldsymbol{x}_t, t)\mathrm{d}\eta
\end{aligned} \tag{21}$$

According to the mean value theorem, there exists some $\tau \in [t, t_{\lambda_\eta}]$ satisfying

$$\|\boldsymbol{h}_\phi(\boldsymbol{x}_{t_{\lambda_\eta}}, t_{\lambda_\eta}) - \boldsymbol{h}_\phi(\boldsymbol{x}_t, t)\|_2 \le (\eta - \eta_t)\left\|\frac{\mathrm{d}\boldsymbol{h}_\phi(\boldsymbol{x}_\tau, \tau)}{\mathrm{d}\eta_\tau}\right\|_2 \tag{22}$$

Besides, the derivative $\frac{\mathrm{d}\boldsymbol{h}_\phi}{\mathrm{d}\eta}$ can be calculated as

$$\begin{aligned}
\frac{\mathrm{d}\boldsymbol{h}_\phi(\boldsymbol{x}_\tau, \tau)}{\mathrm{d}\eta_\tau} &= \frac{\mathrm{d}\boldsymbol{h}_\phi(\boldsymbol{x}_\tau, \tau)}{\mathrm{d}\lambda_\tau}\frac{1}{\mathrm{d}\eta_\tau/\mathrm{d}\lambda_\tau} \\
&= \left(\frac{1}{S_\tau}\frac{\mathrm{d}\boldsymbol{g}_\phi(\boldsymbol{x}_\tau, \tau)}{\mathrm{d}\lambda_\tau} - \frac{\mathrm{d}\log S_\tau}{\mathrm{d}\lambda_\tau}\frac{\boldsymbol{g}_\phi(\boldsymbol{x}_\tau, \tau)}{S_\tau}\right)\frac{1}{L_\tau S_\tau} \\
&= \frac{\frac{\mathrm{d}\boldsymbol{g}_\phi(\boldsymbol{x}_\tau, \tau)}{\mathrm{d}\lambda_\tau} - s_\tau\boldsymbol{g}_\phi(\boldsymbol{x}_\tau, \tau)}{L_\tau S_\tau^2}
\end{aligned} \tag{23}$$

where we have used $\frac{\mathrm{d}\log S_\tau}{\mathrm{d}\lambda_\tau} = s_\tau$ and $\frac{\mathrm{d}\eta_\tau}{\mathrm{d}\lambda_\tau} = L_\tau S_\tau$. Therefore,

$$\begin{aligned}
\|\boldsymbol{D}_{\theta^*}(\boldsymbol{x}_t, t, s) - \boldsymbol{D}_\phi(\boldsymbol{x}_t, t)\|_2 &\le \frac{S_t}{\eta_s - \eta_t}\max_{s \le \tau \le t}\left\|\frac{\mathrm{d}\boldsymbol{g}_\phi(\boldsymbol{x}_\tau, \tau)}{\mathrm{d}\lambda_\tau} - s_\tau\boldsymbol{g}_\phi(\boldsymbol{x}_\tau, \tau)\right\|\int_{\eta_t}^{\eta_s}\frac{\eta - \eta_t}{L_\tau S_\tau^2}\mathrm{d}\eta \\
&\le \max_{s \le \tau \le t}\left\|\frac{\mathrm{d}\boldsymbol{g}_\phi(\boldsymbol{x}_\tau, \tau)}{\mathrm{d}\lambda_\tau} - s_\tau\boldsymbol{g}_\phi(\boldsymbol{x}_\tau, \tau)\right\|\int_{\lambda_t}^{\lambda_s}\frac{L_{t_\lambda}S_{t_\lambda}S_t}{L_\tau S_\tau^2}\mathrm{d}\lambda
\end{aligned} \tag{24}$$

Since we assumed $|l_t|, |s_t| \le c$, according to $\tau \in [t, t_\lambda]$, we have

$$\frac{L_{t_\lambda}}{L_\tau} = e^{\int_{\lambda_\tau}^{\lambda} l_{t'_\lambda}\mathrm{d}\lambda'} \le e^{c(\lambda - \lambda_t)}, \quad \frac{S_{t_\lambda}}{S_\tau} \le e^{c(\lambda - \lambda_t)}, \quad \frac{S_t}{S_\tau} \le e^{c(\lambda - \lambda_t)} \tag{25}$$

Therefore,

$$\int_{\lambda_t}^{\lambda_s}\frac{L_{t_\lambda}S_{t_\lambda}S_t}{L_\tau S_\tau^2}\mathrm{d}\lambda \le \int_{\lambda_t}^{\lambda_s} e^{3c(\lambda - \lambda_t)}\mathrm{d}\lambda = \frac{e^{3c(\lambda_s - \lambda_t)} - 1}{3c} = \frac{(t/s)^{3c} - 1}{3c} \tag{26}$$

Substituting Eqn. (26) in Eqn. (24) completes the proof. $\qquad\square$

Table 3: Experimental configurations.

| Configuration | CIFAR-10 | | FFHQ 64×64 | ImageNet 64×64 |
| --- | --- | --- | --- | --- |
| | Uncond | Cond | Uncond | Cond |
| Learning rate | 0.0004 | 0.0004 | 0.0004 | 0.0004 |
| Student's stop-grad EMA parameter | 0.999 | 0.999 | 0.999 | 0.999 |
| $N$ | 18 | 18 | 18 | 40 |
| ODE solver | Heun | Heun | Heun | Heun |
| Max. ODE steps | 17 | 17 | 17 | 20 |
| EMA decay rate | 0.999 | 0.999 | 0.999 | 0.999 |
| Training iterations | 200K | 150K | 150K | 60K |
| Mixed-Precision (FP16) | True | True | True | True |
| Batch size | 256 | 512 | 256 | 2048 |
| Number of GPUs | 4 | 8 | 8 | 32 |
| Training Time (A800 Hours) | 490 | 735 | 900 | 6400 |

## B  EXPERIMENT DETAILS

### B.1  COEFFICIENTS COMPUTING

At every time $t$, the parameters $l_t$ and $s_t$ can be directly computed according to Eqn. (15) and Eqn. (17), relying solely on the teacher denoiser model $\boldsymbol{D}_\phi$. The computation of $l_t$ involves evaluating $\mathrm{tr}(\nabla_{\boldsymbol{x}_t}\boldsymbol{D}_\phi(\boldsymbol{x}_t, t))$, which is the trace of a Jacobian matrix. Utilizing Hutchinson's trace estimator, it can be unbiasedly estimated as $\frac{1}{N}\sum_{n=1}^{N}\boldsymbol{v}^\top\nabla_{\boldsymbol{x}_t}\boldsymbol{D}_\phi(\boldsymbol{x}_t, t)\boldsymbol{v}$, where $\boldsymbol{v}$ obeys a $d$-dimensional distribution with zero mean and unit covariance. Thus, only the Jacobian-vector product (JVP) $\boldsymbol{D}_\phi(\boldsymbol{x}_t, t)\boldsymbol{v}$ is required, achievable in $\mathcal{O}(d)$ computational cost via automatic differentiation. Once $l_t$ is obtained, the function $\boldsymbol{g}_\phi(\boldsymbol{x}_t, t) = \boldsymbol{D}_\phi(\boldsymbol{x}_t, t) - (1 - l_t)\boldsymbol{x}_t$ is determined. The computation of $s_t$ involves evaluating $\frac{\mathrm{d}\boldsymbol{g}_\phi(\boldsymbol{x}_t, t)}{\mathrm{d}\lambda_t}$, which expands as follows:

$$\begin{aligned}\frac{\mathrm{d}\boldsymbol{g}_\phi(\boldsymbol{x}_t, t)}{\mathrm{d}\lambda_t} &= \frac{\mathrm{d}\boldsymbol{D}_\phi(\boldsymbol{x}_t, t)}{\mathrm{d}\lambda_t} + \frac{\mathrm{d}l_t}{\mathrm{d}\lambda_t}\boldsymbol{x}_t - (1 - l_t)\frac{\mathrm{d}\boldsymbol{x}_t}{\mathrm{d}\lambda_t} \\ &= \frac{\mathrm{d}\boldsymbol{D}_\phi(\boldsymbol{x}_t, t)}{\mathrm{d}\lambda_t} + \frac{\mathrm{d}l_t}{\mathrm{d}\lambda_t}\boldsymbol{x}_t - (1 - l_t)(\boldsymbol{D}_\phi(\boldsymbol{x}_t, t) - \boldsymbol{x}_t)\end{aligned} \tag{27}$$

where $\frac{\mathrm{d}\boldsymbol{D}_\phi(\boldsymbol{x}_t, t)}{\mathrm{d}\lambda_t}$ can also be calculated in $\mathcal{O}(d)$ time by by automatic differentiation.

For the CIFAR-10 and FFHQ 64×64 datasets, we compute $l_t$ and $s_t$ across 120 discrete timesteps uniformly distributed in log space, with 4096 samples used to estimate the expectation $\mathbb{E}_{q(\boldsymbol{x}_t)}$. For ImageNet 64×64, computations are performed across 160 discretized timesteps following EDM's scheduling $(t_{\max}^{1/\rho} + \frac{i}{N}(t_{\min}^{1/\rho} - t_{\max}^{1/\rho}))^\rho$, using 1024 samples to estimate the expectation $\mathbb{E}_{q(\boldsymbol{x}_t)}$. The total computation times for CIFAR-10, FFHQ 64×64, and ImageNet 64×64 on 8 NVIDIA A800 GPU cards are approximately 38 minutes, 54 minutes, and 38 minutes, respectively.

### B.2  TRAINING DETAILS

Throughout the experiments, we follow the training procedures of CTMs. The teacher models are the pretrained diffusion models on the corresponding dataset, provided by EDM. The network architecture of the student models mirrors that of their respective teachers, with the addition of a time-conditioning variable $s$ as input. Training of the student models involves minimizing the consistency loss outlined in Eqn. 7 and the denoising score matching loss $\mathbb{E}_t\mathbb{E}_{p_{\mathrm{data}}(\boldsymbol{x}_0)q(\boldsymbol{x}_t|\boldsymbol{x}_0)}[w(t)\|\boldsymbol{D}_\theta(\boldsymbol{x}_t, t, t) - \boldsymbol{x}_0\|_2^2]$. For the consistency loss, we use LPIPS (Zhang et al., 2018) as the distance metric $d(\cdot, \cdot)$, which is also the choice of CMs. $t$ and $s$ in the consistency loss are chosen from $N$ discretized timesteps determined by EDM's scheduling $(t_{\max}^{1/\rho} + \frac{i}{N}(t_{\min}^{1/\rho} - t_{\max}^{1/\rho}))^\rho$. The Heun sampler in EDM is employed as the solver in Eqn. 7. The number of sampling steps, determined by the gap between $t$ and $s$, is restricted to avoid excessive training time. For CIFAR-10 and FFHQ 64×64, we select $N = 18$ and the maximum number of sampling steps as 17, i.e., not restricting the range of jumping from $t$ to $s$. For ImageNet 64×64, we set $N = 40$ and the maximum number of sampling steps to 20, so that the jumping range is at most half

of the trajectory length. $\text{sg}(\theta)$ in Eqn. 7 is an exponential moving average stop-gradient version of $\theta$, updated by

$$\text{sg}(\theta) = \text{stop-gradient}(\mu\text{sg}(\theta) + (1 - \mu)\theta) \tag{28}$$

We follow the hyperparameters used in EDM, setting $\sigma_{\min} = \epsilon = 0.002, \sigma_{\max} = T = 80.0, \sigma_{\text{data}} = 0.5$ and $\rho = 7$. The training configurations are summarized in Table 3.

We run the experiments on a cluster of NVIDIA A800 GPU cards. For CIFAR-10 (unconditional), we train the model with a batch size of 256 for 200K iterations, which takes 5 days on 4 GPU cards. For CIFAR-10 (conditional), we train the model with a batch size of 512 for 150K iterations, which takes 4 days on 8 GPU cards. For FFHQ 64×64 (unconditional), we train the model with a batch size of 256 for 150K iterations, which takes 5 days on 8 GPU cards. For ImageNet 64×64 (conditional), we train the model with a batch size of 2048 for 60K iterations, which takes 8 days on 32 GPU cards.

### B.3 EVALUATION DETAILS

For single-step as well as multi-step sampling of CTMs, we utilize their deterministic sampling procedure by jumping along a set of discrete timesteps $T = t_0 \rightarrow t_1 \rightarrow \ldots t_{N-1} \rightarrow t_N = \epsilon$ with the consistency function, formulated as the updating rule $\boldsymbol{x}_{t_n} = \boldsymbol{f}_\theta(\boldsymbol{x}_{t_{n-1}}, t_{n-1}, t_n)$. The timesteps $\{t_i\}_{i=0}^N$ are distributed according to EDM's scheduling $(t_{\max}^{1/\rho} + \frac{i}{N}(t_{\min}^{1/\rho} - t_{\max}^{1/\rho}))^\rho$, where $t_{\min} = \epsilon, t_{\max} = T$. We generate 50K random samples with the same seed and report the FID on them.

### B.4 LICENSE

Table 4: The used datasets, codes and their licenses.

| Name | URL | Citation | License |
|------|-----|----------|---------|
| CIFAR-10 | `https://www.cs.toronto.edu/~kriz/cifar.html` | (Krizhevsky et al., 2009) | \ |
| FFHQ | `https://github.com/NVlabs/ffhq-dataset` | (Karras et al., 2019) | CC BY-NC-SA 4.0 |
| ImageNet | `https://www.image-net.org` | (Deng et al., 2009) | \ |
| EDM | `https://github.com/NVlabs/edm` | (Karras et al., 2022) | CC BY-NC-SA 4.0 |
| CM | `https://github.com/openai/consistency_models_cifar10` | (Song et al., 2023) | Apache-2.0 |
| CTM | `https://github.com/sony/ctm` | (Kim et al., 2023) | MIT |

We list the used datasets, codes and their licenses in Table 4.

## C ADDITIONAL SAMPLES

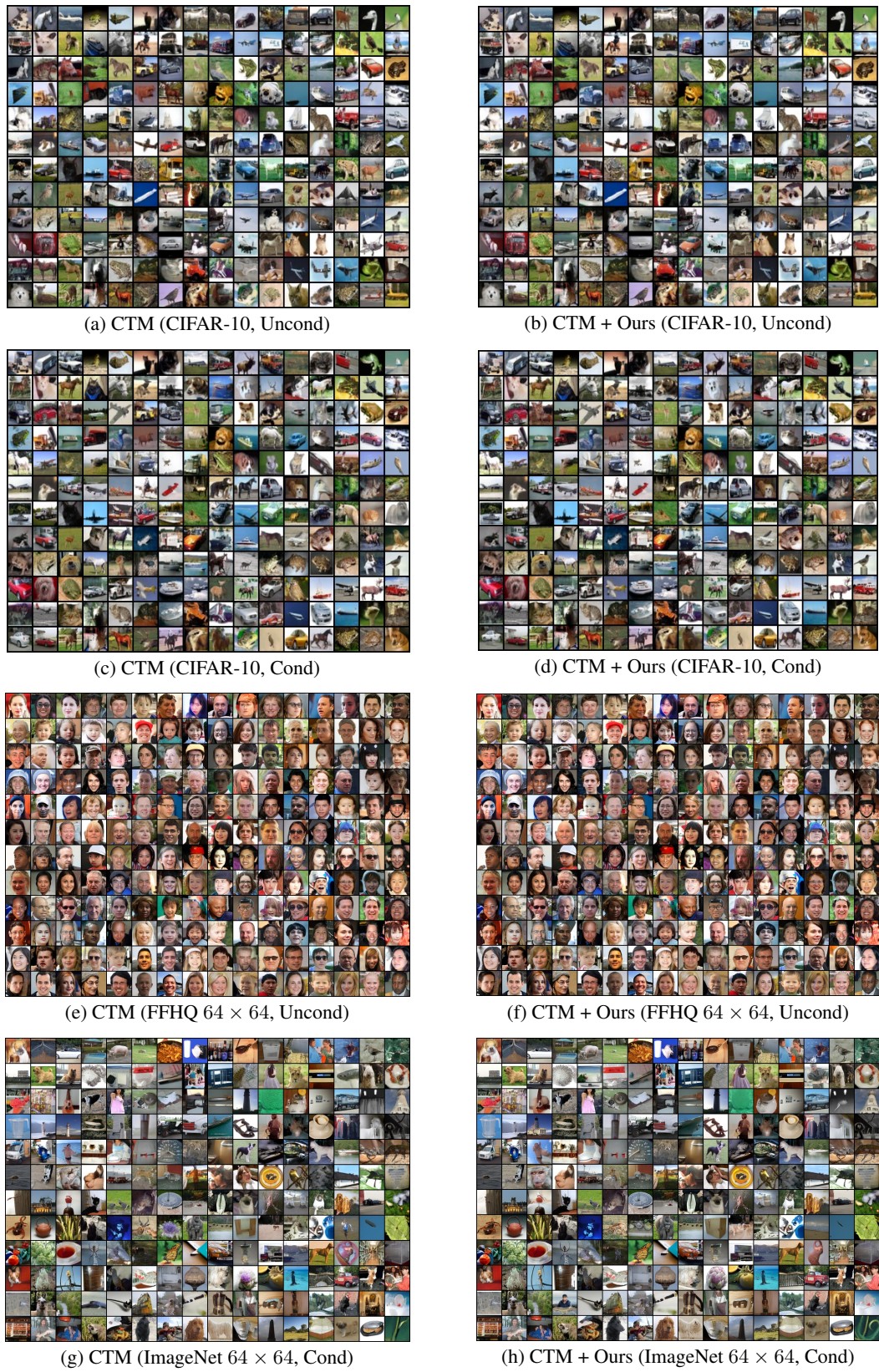

(a) CTM (CIFAR-10, Uncond)

(b) CTM + Ours (CIFAR-10, Uncond)

(c) CTM (CIFAR-10, Cond)

(d) CTM + Ours (CIFAR-10, Cond)

(e) CTM (FFHQ $64 \times 64$, Uncond)

(f) CTM + Ours (FFHQ $64 \times 64$, Uncond)

(g) CTM (ImageNet $64 \times 64$, Cond)

(h) CTM + Ours (ImageNet $64 \times 64$, Cond)

Figure 8: Random samples produced by CTM and CTM + Analytic-Precond (Ours) with NFE=2.

