# OpenReview forum: "Elucidating the Preconditioning in Consistency Distillation"
_ICLR.cc/2025/Conference — ICLR 2025 Poster_

### Official Review · Reviewer_TK51 · 2024-10-31

**Soundness:** 3
**Presentation:** 3
**Contribution:** 2
**Rating:** 6
**Confidence:** 3

**Summary:**

Consistency (trajectory) distillation typically uses a network parameterized as $f(x, t, s) = \alpha_{t, s} F_\theta(x, t, s) + \beta_{t, s} x$, with specific expressions for the coefficient (i.e. preconditioners) $\alpha_{t, s}, \beta_{t, s}$ so that the boundary conditions are automatically satisfied. This paper proposes an efficient method to find alternative preconditioners that yield improved performance and faster training. They derive the expressions for the preconditioners by re-expressing the underlying ODE in terms using certain additional variables and propose simple objectives whose minimizers can be found analytically to set the values for these variables.

**Strengths:**

Consistency (trajectory) distillation represents an extremely popular approach for fast generation, solving the main drawback of diffusion models. Finding approaches to improve distillation, either the final performance achieved or its training cost, is extremely relevant to improve the practicality and applicability of large models currently being trained in multiple domains.

This paper tackles this problem by changing the preconditioning parameters used for the neural network. To my knowledge the procedure described in the paper is novel, and empirical results show that, for CTMs, the proposed approach yields a training speedup of 2x for multiple datasets.

The final proposed method is quite simple to implement, with closed-form expressions for the preconditioners. (However, these expressions involve nonlinear functions intractable expectations which are estimated with samples, meaning that the actual values obtained have both bias and variance.)

**Weaknesses:**

The paper deals with the consistency (trajectory) distillation case. In practice, a very relevant alternative that is also widely used (and also uses similar preconditioners for the network) is consistency training. That is, directly training fast samplers without distilling a base model. The paper does not address this problem at all. While I understand this is not necessary, and distillation in itself is an important task, it would be interesting to see whether certain ideas from this work can be extended to consistency training. (Most of the expressions for the preconditioners rely on having a pre-trained model, so unclear how to generalize this approach, if possible.)

The approach seems to help when using CTM with 2 steps or more. For single step generation performance and training curves pretty much overlap with and without the proposed approach. 1 step generation plays an important role for real time generation, and it would be very interesting to develop methods that can improve training and final performance in that setting as well. Additionally, the fact that the derived preconditioners are essentially the same as the ones naively used by consistency (trajectory) distillation raises the question of whether there are other preconditioners that can be used that might help in this case as well. Again, not exploring this does not reduce the paper’s merit, but I think it is an interesting question too.

The method does not yield any benefits when used in concert with a GAN auxiliary loss. Using this loss has been observed to lead to improved performance, and indeed, the best results reported in the paper are using the GAN loss, if I understand correctly. The proposed approach does not yield benefits (but at least does not hurt) when using this auxiliary loss.

**Questions:**

How does performance and the actual values for the preconditioners change as we change the number of samples used to estimate Eqs. 15 and 17 (expressions for the preconditioners)? Results in the paper are obtained estimating them for 120 different times, using 4096 samples to estimate expectations. This yields good results for 2 step sampling, at a modest computational cost. This is definitely not strictly necessary, but I think it could be interesting to see how the performance and preconditioner values change as we change the number of samples used. The estimators used have variance and bias, both of which decrease with the number of samples. How small can we make the number of samples used while still retaining good performance? Can we further boost performance using more samples?

This is all for distillation. Did you think about consistency training? These preconditioning parameters also show up in that case, but we don’t have access to the teacher, so it is unclear how to use the ideas in this work.

While multi-step consistency models have been observed to underperform CTMs, it would be nice to have values in some of the results reported. I understand training is exactly the same, so I’m not expecting the new preconditioners to help there. But it would be nice having plain multi-step CMs in some results.

What dataset is used for the GAN experiment?

---

> ### Author Response · Authors · 2024-11-23
>
> Thank you for your time and effort in reviewing our work! We appreciate your thoughtful and positive feedback on our work. We hope the responses below address your concerns.
>
> *Q1: How does performance and the actual values for the preconditioners change as we change the number of samples used to estimate Eqs. 15 and 17 (expressions for the preconditioners)? How small can we make the number of samples used while still retaining good performance? Can we further boost performance using more samples?*
>
> **A**: We did not extensively tune the number of samples due to the high cost of training experiments. Furthermore, even with the same number of samples, the trace estimator's variance and the specific dataset partition used can lead to differing estimations. Based on our experience, selecting a sample size between 1024 and 4096 produces preconditioners that appear visually similar and generally achieve speed-ups exceeding 1.5x. However, using fewer than 200 samples introduces estimation bias and degrades performance. We also experimented with larger sample sizes, up to 8192, but observed no additional improvements. This suggests that we have exploited the potential improvement space of the preconditioning, and optimizing the preconditioning alone may have upper bound and cannot bring further benefits.
>
> *Q2: Did you think about consistency training? These preconditioning parameters also show up in that case, but we don’t have access to the teacher, so it is unclear how to use the ideas in this work.*
>
> **A**: We believe consistency distillation is a more efficient and promising approach, as consistency training from scratch typically requires more iterations and results in sub-par performance. Intuitively, diffusion models can be stably and fastly trained to give the tangent of the ODE trajectory. It averages the high-variance tangents from random data-noise pairs and offers stable predictions. Even for consistency training, preparing a diffusion models in advance may also be beneficial. For example, in Appendix B.3 in the consistency model paper, they claim that they initialize the consistency model with the pretrained diffusion model when conducting continuous-time consistency training, and this initialization significantly stabilize the training.
>
> *Q3: While multi-step consistency models have been observed to underperform CTMs, it would be nice to have values in some of the results reported.*
>
> **A**: Thanks for your suggestion. We additionally tested multi-step CM and presents the results below.
>
> |NFE|1|2|3|5|8|10|
> |:--------|:-------:|:-------:|:------:|:------:|:-------:|:-------:|
> |CM|3.54|2.94|2.93|2.95|3.22|3.30|
> |CTM|3.57|3.00|2.82|2.59|2.67|2.56|
> |CTM+Ours|3.57|2.92|2.75|2.62|2.50|2.51|
>
> As expected, CM cannot boost its performance with more sampling steps. It needs to perform alternative backward and forward jumps. As the forward jump is stochastic, it no longer ensures trajectory consistency and instead accumulates error.
>
> *Q4: What dataset is used for the GAN experiment?*
>
> **A**: CIFAR-10 is used as other datasets are in 64x64 resolution and demand higher computational resources. We have revised the paper to specify.

---

> > ### Comment · Reviewer_TK51 · 2024-11-26
> > **Thanks for your response**
> >
> > Thanks for your responses. I keep my acceptance score.
> >
> > Just one comment regarding "We believe consistency distillation is a more efficient and promising approach, as consistency training from scratch typically requires more iterations and results in sub-par performance." I agree with the performance gap. Consistency training has some benefits though, specially for very large models where even keeping two copies of the model may be problematic. Though I don't think not dealing with this problem takes merit away from the paper; distillation is also a very important topic by itself.

---

> > > ### Author Response · Authors · 2024-11-26
> > > **Thank you**
> > >
> > > Thank you for your thoughtful feedback and for keeping your acceptance score. We agree that consistency training can have benefits in certain scenarios and will leave improvements on them for further research.

---

### Official Review · Reviewer_yEcg · 2024-11-04

**Soundness:** 3
**Presentation:** 2
**Contribution:** 3
**Rating:** 6
**Confidence:** 2

**Summary:**

The paper works on _Preconditioning_, a technique used in the consistency distillation of diffusion models to obtain consistency functions that directly satisfy the boundary conditions required by the problem. Preconditioning consists in linearly linking the input to the output of a network. In the literature, the choice of linear coefficients is based on intuition. The paper introduces instead a new analytical method, called _Analytic-Precond_, for setting the coefficients. The method consists in applying a parametric discretization of the probability flow ODE, and then optimizing the parameters by minimizing the gap between the optimal student and the teacher, while keeping the discretization as robust as possible. Finally, some numerical proofs show that the derived result leads to a speed-up in the inference of diffusion models.

**Strengths:**

The paper presents strong mathematical arguments to support the choice of coefficients, including an explanation for the CMT choices that is not just based on intuition as previous methods.
- The paper shows numerical proofs of the claims made, underlying when _Analytic-Precond_ offers no advantage (single step) and when it does (two or more steps).

**Weaknesses:**

- The paper might be a little hard to read for who is not familiar with distillation. I personally took a while to grasp the setting and all the notation. For example, $\phi$ is used many times before definition. It could be worth having a brief discussion about some nomenclature like _teacher_ & _student_.

**Questions:**

- The paper focuses on the case where $f=0, g=\sqrt{2t}$ because of the recent literature. Is the method still applicable for other choices of $f$ and $g$?
- As far as I understand, the whole discussion depends on finding a good discretization of ODE (2). Both (9) and (13) use first order (Euler) methods. Can we get more insight if we try to use a better integrator?

Minor:
- Why is $q_T$ on line 118 a 0 mean Gaussian? Do we have some condition on $\mathbb{E}[q_0]$?
- The $\lambda_t$ below equation (11) might be confused with $\lambda(t)$ in equations (5), (7).
- x$ and $x_t$ are used interchangeably in the RHS and LHS of the equations, better to be consistent. Example: line 182, line 276.
- Is the code of the experiments released?

---

> ### Author Response · Authors · 2024-11-23
>
> Thank you for your time and effort in reviewing our work! We appreciate your thoughtful and positive feedback on our work. We hope the responses below address your concerns.
>
> *Q1: The paper focuses on the case where $f=0, g=\sqrt{2t}$ because of the recent literature. Is the method still applicable for other choices of $f$ and $g$?*
>
> **A**: Sure! A more convenient way is to represent the forward process defined by the SDE as $q(x_t|x_0)=\mathcal{N}(\alpha_tx_0,\sigma_t^2I)$. The coefficients $\alpha_t,\sigma_t$ are determined by $f(t),g(t)$ in the forward SDE and satisfy $f(t)=\frac{d\log \alpha_t}{d t}$, $g^2(t)=\frac{d \sigma_t^2}{d t}-2\frac{d\log \alpha_t}{d t}\sigma_t^2$ [1]. The case $f=0, g=\sqrt{2t}$ is adopted in the recent literature as it corresponds to $\alpha_t=1,\sigma_t=t$ which is quite simple, and the corresponding diffusion ODE is $\frac{dx_t}{dt}=\frac{x_t-\hat x_0}{t}$, where $\hat x_0$ is the predicted $x_0$ by the denoiser function. For more general $f,g$, or equivalently $\alpha_t,\sigma_t$, we can apply some transformations to turn it into the simple case $\alpha_t=1,\sigma_t=t$.
>
> Specifically, if we define $x_t'=\frac{x_t}{\alpha_t}$, then $x_t'$ satisties the forward process $q(x_t'|x_0')=\mathcal{N}(x_0',\frac{\sigma_t^2}{\alpha_t^2}I)$. If we additionally define a new time $t'=\frac{\sigma_t^2}{\alpha_t^2}$, the noise-to-signal ratio, then the forward process of $x_t'$ corresponds to $\alpha_t=1,\sigma_t=t'$. Intuitively, this creates a "wrapper" for the original process and turn them into the simple case. Therefore, the corresponding diffusion ODE can be written as $\frac{dx_t'}{dt'}=\frac{x_t'-\hat x_0}{t'}$, and we can still follow the procedure in the paper to derive preconditionings.
>
>
> [1] Variational Diffusion Models
>
> *Q2: As far as I understand, the whole discussion depends on finding a good discretization of ODE (2). Both (9) and (13) use first order (Euler) methods. Can we get more insight if we try to use a better integrator?*
>
> **A**: That is an insightful understanding and question. We think in the design of preconditionings, we can only rely on first-order Euler methods. Better integrators such as high-order runge-kutta methods, require multiple evaluations of the ODE drift (which involves the network) to perform a single ODE step. However, in the case of preconditioning, which is a linear combination of $x_t$ and the network output, it will be much more expensive if we combine multiple network outputs, considering that they are involved in the gradient backpropagation during training.
>
> *Q3: Why is $q_T$ on line 118 a 0 mean Gaussian? Do we have some condition on $\mathbb{E}[q_0]$?*
>
> **A**: The zero-mean is only an approximation. There is no special restriction on the data distribution $q_0$. The intuition is, in practice, the data range is small (normalized to [-1,1]), while the final time $T$ is set to a very large value (like 80). This is often called a "variance-exploding" noise schedule. Therefore, compared to the large variance of the Gaussian distribution, the mean is relatively negligble. This can be understood from another perspective. The noisy data $x_t=x_0+t\epsilon,\epsilon\sim\mathcal{N}(0,I)$ has a very large scale when $t$ is large. For stability, before input into the network, it will be firstly normalized to something like $\frac{x_t}{\sqrt{1+t^2}}=\frac{1}{\sqrt{1+t^2}}x_0+\frac{t}{\sqrt{1+t^2}}\epsilon$. Therefore, as $t\rightarrow\infty$, the component of $x_0$ will tend to 0.
>
> *Q4: The $\lambda_t$ below equation (11) might be confused with $\lambda(t)$ in equations (5), (7).*
>
> **A**: Thank you for your suggestion. We have revised the notation of the weighting function from $\lambda(t)$ to $w(t)$.
>
> *Q5: $x$ and $x_t$ are used interchangeably in the RHS and LHS of the equations, better to be consistent. Example: line 182, line 276.*
>
> **A**: Thanks for spotting the typos. We have fixed them in the revised paper.
>
> *Q6: Is the code of the experiments released?*
>
> **A**: Due to the high cost of the training experiments and the need for proper permissions, we plan to release the code upon acceptance.

---

> > ### Comment · Reviewer_yEcg · 2024-11-26
> >
> > I thank the authors for the detaild answer, I apriciate all the clarification provided. As a non-expert of the subject, I keep my acceptance score since I think the paper should be accepted, but I defer to other reviewers a discussion of the relevance of the content.

---

> > > ### Author Response · Authors · 2024-11-26
> > > **Thank you**
> > >
> > > We are glad to know our responses help. Thank you for your support and transparency.

---

### Official Review · Reviewer_6jeK · 2024-11-05

**Soundness:** 3
**Presentation:** 3
**Contribution:** 3
**Rating:** 6
**Confidence:** 3

**Summary:**

The paper titled "Elucidating the Preconditioning in Consistency Distillation" examines consistency distillation techniques for diffusion models, where a student model learns to follow the probability flow trajectory set by a teacher model. This distillation accelerates generation by reducing the inference steps. The paper specifically explores preconditioning, a method that combines input data with network outputs to improve stability during training. Traditionally, preconditioning has been handcrafted, but this paper introduces a theoretically optimized method named "Analytic-Precond." This new approach minimizes the gap between teacher and student denoisers, thereby improving training efficiency and trajectory alignment. Experimental results demonstrate that Analytic-Precond achieves up to 3x acceleration in training across various datasets, indicating its potential in enhancing consistency models for faster multi-step generation.

**Strengths:**

**Theoretical Innovation in Preconditioning**: The paper introduces "Analytic-Precond," a novel, analytically derived preconditioning method that theoretically optimizes the consistency distillation process. This goes beyond prior handcrafted preconditionings, offering a principled approach that minimizes the consistency gap between the teacher and student models. This theoretical grounding not only strengthens the methodology but also provides new insights into consistency distillation.

**Significant Training Acceleration**: Experimental results show that Analytic-Precond achieves 2-3x faster training in multi-step generation tasks on standard datasets. This improvement in speed is impactful, especially for resource-intensive applications of diffusion models, as it directly addresses the bottleneck of slow inference that has historically limited diffusion models.

**Weaknesses:**

- This paper does not provide whether BCM is better than CTM+ Analytic-Precond in terms of FID.
- Analytic-Precond does not perform better when GAN is incorporated into the CTM. Can the authors provide an explanation or intuition for this?

**Questions:**

See weakness

---

> ### Author Response · Authors · 2024-11-23
>
> Thank you for your time and effort in reviewing our work! We appreciate your thoughtful and positive feedback on our work. We hope the responses below address your concerns.
>
> *W1: This paper does not provide whether BCM is better than CTM+ Analytic-Precond in terms of FID.*
>
> **A**: CTM, even combined with our Analytic-Precond, is slightly worse than BCM in terms of FID. The reason is that BCM adopts techniques from improved Consistency Training (iCT), such as better scheduler function and reweighting function. As our work mainly focuses on preconditioning instead of other techniques, we demonstrate in Figure 5 that BCM's preconditioning is not superior and cannot bring improvements to CTM, while ours can.
>
> *W2: Analytic-Precond does not perform better when GAN is incorporated into the CTM. Can the authors provide an explanation or intuition for this?*
>
> **A**: Sure! The analyses of preconditioning in our work, such as the consistency gap, are built on the investigations of how to learn better trajectory jumpers and maintain fidelity to the teacher ODE trajectory. However, the incorporation of the GAN loss is merely to enhance the FID at 1-step. As shown in Figure 6, in this scenario, the consistency function no longer faithfully adheres to the teacher ODE trajectory, and one-step generation is even better than two-step, deviating from our theoretical foundations.

---

> > ### Comment · Reviewer_6jeK · 2024-11-23
> >
> > Thanks for your comments. However, you said "CTM, even combined with our Analytic-Precond, is slightly worse than BCM in terms of FID", which means that BCM's better performance does not come from the preconditioning. Therefore, I wonder (1) is adjusting preconditioning the most efficient way to improve the performance? (2) if you combine BCM with your precoditioning methods, will it be better than BCM? If the answer to (2) is yes, I think I will give score 8. I will revise the score to 6 for now.

---

> > > ### Author Response · Authors · 2024-11-27
> > >
> > > Thank you for your feedback. Though our method may not be the most efficient way, we believe it is universal for enhancing trajectory consistency and can be combined with other techniques. As BCM did not provide code on CIFAR10, we refer to and adapt their ImageNet64 version for distillation. We provide some preliminary result on 2-step FID.
> > >
> > > |Iteration|10k|20k|30k|40k|50k|
> > > |:--------|:-------:|:-------:|:------:|:------:|:-------:|
> > > |BCM|3.71|3.38|3.23|3.10|3.05|
> > > |BCM+Ours|3.47|3.26|3.12|3.02|2.99|
> > >
> > > Despite potential implementation differences, this can serve as evidence of the applicability of our method.

---

### Official Review · Reviewer_ae52 · 2024-11-06

**Soundness:** 3
**Presentation:** 3
**Contribution:** 2
**Rating:** 6
**Confidence:** 3

**Summary:**

This paper proposed a general paradigm of preconditioning design in consistency distillation, which is a common technique used for accelerating the inference time of consistency models based on teacher-student training (i.e., knowledge distillation). Specifically, this paper focused on preconditioning, which is a vital technique for stabilizing consistency distillation. Compared to previous hand-crafted choices of preconditioning, this paper proposed a principled way called "Analytic-Precond" to analytically optimize the preconditioning based on the consistency gap associated with the teacher probability flow ODE. Numerical experiments on multiple datasets are included to justify the effectiveness of "Analytic-Precond".

**Strengths:**

1. Complete proofs are included for each proposition in the manuscript.
2. Extensive numerical experiments are provided to validate the effectiveness of the proposed methodology.

**Weaknesses:**

Presentation of the manuscript can be further improved by rewriting certain phrases and expanding on some technical details. For instance, the phrase "CMs aim to a consistency function" on line 134 might be better rephrased as "CMs aim to learn a consistency function". For possible ways of explaining technical details in a better way, one may refer to the "Questions" section below.

**Questions:**

From line 265-266 the authors mentioned that the parameter $l_t$ in "Analytic-Precond" is chosen to be the minimizer of the expected gradient norm $E_{q(x_t)}[\|\nabla_{x_t}g_{\phi}(x_t,t)\|_F]$ based on earlier work [1]. Would it be possible for the authors to further expand on why such choice ensures the robustness of the resulting ODE again errors in $x_t$? Which section/part of [1] discussed the reason behind such choice?

References:

[1] Zheng, Kaiwen, Cheng Lu, Jianfei Chen, and Jun Zhu. "Improved techniques for maximum likelihood estimation for diffusion odes." In International Conference on Machine Learning, pp. 42363-42389. PMLR, 2023.

---

> ### Author Response · Authors · 2024-11-23
>
> Thank you for your time and effort in reviewing our work! We appreciate your thoughtful and positive feedback on our work. We hope the responses below address your concerns.
>
> *W1: For instance, the phrase "CMs aim to a consistency function" on line 134 might be better rephrased as "CMs aim to learn a consistency function"*
>
> **A**: Thank you for carefully reading our paper and pointing out the typo. We have fixed it in the revised paper.
>
> *Q1: Would it be possible for the authors to further expand on why such choice ensures the robustness of the resulting ODE again errors in $x_t$?*
>
> **A**: Sure! The reference you mentioned actually points to another paper that applies Rosenbrock-type exponential integrators to diffusion ODEs. We would like to give an illustrative example to explain its core idea and how it enhances the robustness against errors in $x_t$.
>
> Suppose we want to solve an ODE from $t_n$ to $t_{n+1}$, where $t_{n+1}>t_n$:
>
> $$
> \frac{dx_t}{dt}=F(x_t)
> $$
>
> The core idea of Rosenbrock-type exponential integrators is to separate as much linear component from $F$, as the linear part can be analytically absorbed into a "modulation" of the ODE. Specifically, let $F(x_t)=-l x_t+N(x_t)$, where $l>0$ (so that the ODE is not explosive in forward time), and the non-linear part $N$ satisfies that $\frac{dN(x_t)}{dx_t}\approx 0$ at $t=t_n$. Then by the chain rule, the original ODE can be turned into an ODE that describes the evolution of $e^{l t}x_t$ instead of $x_t$.
>
> $$
> \frac{d(e^{l t}x_t)}{dt}=e^{l t}\frac{dx_t}{dt}+l e^{l t}x_t=e^{l t}(F(x_t)+l x_t)=e^{l t}N(x_t)
> $$
>
> Denote $h=t_{n+1}-t_n$, the Euler discretization of the original ODE is
> $$
> x_{n+1}=h(-l x_n+N(x_n))\Rightarrow x_{n+1}=(1-hl)x_n+hN(x_n)
> $$
>
> The Euler discretization of the modulated ODE is
> $$
> e^{l t_{n+1}}x_{n+1}-e^{l t_{n}}x_{n}=he^{l t_{n}}N(x_{n})\Rightarrow x_{n+1}=e^{-l h}(x_n+hN(x_n))
> $$
>
> Suppose $x_n'=x_n+e_n$ is the perturbed $x_n$ with error $e_n$, and $e_{n+1}=x_{n+1}'-x_{n+1}$ is the resulting error in $x_{n+1}$. As $\frac{dN(x)}{dx}\approx 0$ at $x=x_n$, we can omit $N(x_n')-N(x_n)$ for small $e_n$. Therefore, $|e_{n+1}|=|1-hl||e_n|$ for the original ODE, which may amplify the error when $h$ is large and $|1-hl|>1$. Instead, $|e_{n+1}|=e^{-l h}|e_n|<|e_n|$ after separating the linear part and modulate the ODE.

---

> > ### Comment · Reviewer_ae52 · 2024-11-28
> >
> > Thank you so much for your clarification - I would like to keep my acceptance score.

---

> > > ### Author Response · Authors · 2024-11-28
> > > **Thank you**
> > >
> > > We are glad our clarification is helpful and appreciate your support.

---

### Meta-Review · Area_Chair_FdrL · 2024-12-19

**Metareview:**

This paper studies the design criteria of the preconditioning in consistency distillation and propose a novel and principled preconditioning with speedup 2x to 3x. Compared to previous hand-crafted choices of preconditioning, this submission developed a principled way called "Analytic-Precond" to analytically optimize the preconditioning based on the consistency gap associated with the teacher probability flow ODE. Experiments on several datasets justify the effectivenss of the proposed method. The AC recommend the authors to include the reviewers' feedback and suggestions.

**Additional Comments On Reviewer Discussion:**

During the discussion, the authors addressed the issues raised by the reviewers. For example, the robustness of ODE against errors under the selected choice, combining BCM/CTM with the precoditioning methods,

---

### Decision · Program_Chairs · 2025-01-22

Accept (Poster)